# Molecular Characterization and Epidemiology of Human Noroviruses in the Sverdlovsk Region, Russian Federation

**DOI:** 10.3390/v17091243

**Published:** 2025-09-15

**Authors:** Roman Bykov, Tarek Itani, Daria Pletenchuk, Olesia Ohlopkova, Alexey Moshkin, Marina Stepanyuk, Aleksandr Semenov

**Affiliations:** 1Federal Budgetary Institution of Science, «Federal Scientific Research Institute of Viral Infections «Virome»», Federal Service for Surveillance on Consumer Rights Protection and Human Wellbeing, Ekaterinburg 620030, Russia; itani_tm@niivirom.ru (T.I.);; 2Research Institute of Virology, Federal State Budgetary Scientific Institution «Federal Research Center for Fundamental and Translational Medicine», Novosibirsk 630060, Russia; 3Department of Medical Microbiology and Clinical Laboratory Diagnostics, Ural State Medical University, Ekaterinburg 620109, Russia; 4Institute of Natural Sciences and Mathematics, Ural Federal University Named After the First President of Russia B.N. Yeltsin, Ekaterinburg 620075, Russia

**Keywords:** next-generation sequencing (NGS), HuNoVs GII.7[P7] and GII.4[P16], phylogenetic analysis, non-synonymous mutations, genotypic composition

## Abstract

Human noroviruses (HuNoVs) stand as the primary cause of acute viral gastroenteritis outbreaks worldwide, particularly impacting children under the age of five. In Russia, reports of norovirus gastroenteritis have surged, especially in the post-COVID-19 era starting in 2022, with elevated infection rates reported into 2024. These viruses exhibit significant mutational variability, leading to the emergence of recombinant strains that can evade immune responses. A comprehensive examination of the complete genome is crucial for understanding the evolution of norovirus genes and for predicting potential outbreaks. This research focuses on analyzing the genotypic composition of HuNoVs circulating in the Sverdlovsk region during 2024, using Sanger sequencing and next-generation sequencing (NGS). Biological samples were collected (n = 384) from patients diagnosed with norovirus infection within the region. Bioinformatics analysis targeted the nucleotide sequences of the ORF1/ORF2 fragment and the assembly of complete genomes for the GII.4 and GII.7 genotypes. In total, 220 HuNoVs were characterized, representing 57.3% of the collected samples. The main capsid variants forming the predominant genotypic profile included GII.4 (n = 88, 40%), GII.7 (n = 86, 39%), and GII.17 (n = 14, 6%). Using NGS, we successfully assembled 8 out of 10 complete genomes for noroviruses GII.4[P16] and GII.7[P7]. Non-synonymous substitutions appeared at amino acid sites corresponding to the subdomains of VP1 in these strains. This molecular–genetic analysis provides contemporary insights into the genotypic composition, circulation patterns, and evolutionary dynamics associated with the dominant genovariants GII.4[P16] and GII.7[P7].

## 1. Introduction

Human noroviruses (HuNoVs) serve as the primary etiological factor behind most outbreaks of acute viral gastroenteritis globally. The global burden from infectious diseases presents a significant challenge for public health, with approximately 655 million infections and 213,515 deaths reported annually in the human population. The highest rates of morbidity and mortality affect children under five years [1,2,3].

In the Russian Federation, there is a continuous trend of increasing morbidity rates from norovirus gastroenteritis, particularly observed in the post-COVID-19 period. According to the published data from the State Report by Rospotrebnadzor, the Federal Service for Surveillance on Consumer Rights Protection and Human Wellbeing, titled “Sanitary and Epidemiological Wellbeing in the Russian Federation in 2024”, the morbidity rate for HuNoVs in Russia reached 37.62 per 100,000 people by the end of 2024, reflecting a 9% increase compared to levels reported in 2023. The rise in HuNoV morbidity can be attributed to the shift in circulating viral genotypes within the Russian Federation from GII.17[P17] in 2023 to GII.7[P7] in 2024, as well as the expanded coverage of laboratory diagnostics for associated acute gastroenteritis (AGE) using molecular methods. Notably, children comprised 82% of the cases that were reported. The highest incidence of HuNoVs occurred among children aged one to two years, with a morbidity rate of 516.12 per 100,000 people. Among the regions of the Russian Federation, the Sverdlovsk region exhibits a concerning epidemiological situation regarding HuNoVs, ranking fourth in overall morbidity with a rate of 126.38 per 100,000 people [4].

HuNoVs belong to the genus *Norovirus*, which is part of the family *Caliciviridae*. Their genome consists of single-stranded positive-sense RNA, which contains three main open reading frames (ORFs) that regulate the synthesis of viral proteins. ORF1 encodes the RNA-dependent RNA polymerase (RdRp) and other proteins involved in the replication cycle. ORF2 encodes the major capsid protein, VP1, while ORF3 encodes the minor capsid protein, VP2 [5,6]. The virus-encoded protein VPg is covalently linked to the 5′ end of the genome, which plays an essential role in the initiation of translation factors in host cells. The 3′ end features a polyadenylated tail. A key function in the penetration of norovirus virion into target cells is performed by the structural protein VP1, composed of a complex with domains, including the N/S domain and P domain [7]. The N/S domain serves as a shell that participates in forming the icosahedral symmetry type for the capsid. The P domain can be conventionally divided into two domains, P1 and P2 [8], which act as antigenic determinants and dictate the binding specificity to polysaccharide complexes on target cells (Figure 1). The norovirus genome exhibits high mutational variability due to recombination events in the regions connecting the two open reading frames, ORF1 and ORF2, as well as within ORF2 itself [9].

Recombinational variability results in a wide variety of recombinant strains that can evade the adaptive immune response in humans. This situation necessitates routine molecular–genetic monitoring and annual updates to the genotypic composition of HuNoVs across various regions in Russia and epidemiologically significant areas in the world with high disease prevalence [9].

To achieve a more objective assessment of divergent evolution events among circulating HuNoV genotypes and to forecast potential future outbreaks, a thorough analysis of the complete norovirus genome is necessary. Several limitations hinder NGS application for HuNoV sequencing such as low viral load and the absence of a sensitive cell culture model for norovirus cultivation, which prevents the acquisition of a high viral RNA concentration after extraction. The above-mentioned limitations present a significant challenge in the implementation of norovirus sequencing using NGS methods; thus, there is a need to develop alternative ways to address the existing difficulties. A viable solution to the identified challenges may be the viral analysis of the whole genome, which amplifies all RNA-containing microorganisms present in the studied sample. Another approach is the development of a specific primer panel to enrich the HuNoV genome. Therefore, this study aims to conduct molecular characterization of the circulating HuNoV genotypes in the Sverdlovsk region in 2024.

## 2. Materials and Methods

### 2.1. Stool Samples

In 2024, fecal sample collection was organized from patients aged between 1 and 23 years, diagnosed with norovirus infection, in municipalities across the Sverdlovsk region showing the highest HuNoV morbidity rates. The collection of stool samples from patients with a confirmed diagnosis of HuNoVs was organized in four municipalities during 2024: Ekaterinburg, Kamensk-Uralsky, Nizhny Tagil, and Sukhoy Log (Figure 2). Samples from patients were screened for HuNoVs by either an enzyme immunoassay (Norovirus-antigen-enzyme immunoassays-Best, Vector-Best, Novosibirsk, Russian Federation) or qPCR (AmpliSens^®^ Norovirus GI-GII-FL, Central Scientific Research Institute of Epidemiology, Moscow, Russian Federation) at the following facilities: The Ekaterinburg Consultative Diagnostic Center and the State Autonomous Healthcare Institution of the Sverdlovsk Region Children’s City Hospital (Kamensk-Uralsky). Only positive samples were submitted to the Federal Scientific Research Institute of Viral Infections’ “Virome” (n = 384) for further molecular analysis.

### 2.2. Ethics Statement

This research was approved by the Local Ethics Committee at the Federal Budgetary Institution of Science “Federal Scientific Research Institute of Viral Infections ‘Virome’ Federal Service for Surveillance on Consumer Rights Protection and Human Wellbeing Rospotrebnadzor” (Protocol Nº 1 dated 17 March 2022). Fecal samples were collected from medical institutions in the Sverdlovsk region: The Ekaterinburg Consultative Diagnostic Center and the State Autonomous Healthcare Institution of the Sverdlovsk region Children’s City Hospital (Kamensk-Uralsky). All patients gave their written consent to participate in the study, and patient data were stored anonymously and securely. All fecal samples were transported to the Federal Scientific Research Institute of Viral Infections “Virome” and stored at −20 °C until analysis.

### 2.3. Isolation of RNA and PCR

A 10% suspension in physiological normal saline was prepared from fecal samples, vortexed, and clarified by centrifugation for 3 min at 10,000× *g*. Nucleic acids were extracted using the Riboprep^®^ kit (Central Research Institute of Epidemiology, Moscow, Russian Federation). Subsequently, reverse transcription was performed to obtain cDNA from RNA templates using the REVERTA-L kit (Central Research Institute of Epidemiology, Moscow, Russian) according to the manufacturer’s protocol. Polymerase chain reaction (PCR) was set up using 5X ScreenMix-HS (EUROGEN, Moscow, Russian) reaction mix, deionized water, and a pair of specific oligonucleotide primers. The selected primer pair flanked a region of the norovirus genome spanning two genetic groups (GI, GII) corresponding to the ORF1/ORF2 region. For GI, the primer pair included MON432 (forward) and GISKR (reverse), while for GII, it comprised the primer pair MON431 (forward) and GIISKR (reverse) at a molar concentration of 20 [pmol/L] [10]. Amplicon electrophoresis was performed on a 2% agarose gel (TBE buffer, 180 V, for 30 min). cDNA purification of the agarose gel was conducted using the Cleanup Standard reagent kit (EUROGEN, Moscow, Russia).

### 2.4. Sequencing a Fragment ORF1/ORF2 of the Norovirus Genome Using the Sanger Method

Target genome fragment amplification was followed by chain termination through the incorporation of dideoxynucleotides utilizing the BrilliantDyeTM Terminator (v3.1) Cycle Sequencing Kit, along with the BrilliantDye buffer and two pairs of oligonucleotide primers (GI—MON432/GISKR; GII—MON431/GIISKR at a molar concentration of 3.2 [pmol/L]) [10]. Norovirus genome fragments were sequenced on the ABI 3130 Genetic Analyzer (Applied Biosystems, Foster City, CA, USA) as previously described.

### 2.5. Preparation of NGS Libraries

Total RNA was extracted from fecal suspensions using the ExtractRNA kit (EUROGEN, Russia). cDNA synthesis followed by PCR was carried out using the RNAscribe RT Reverse Transcriptase kit (Biolabmix, Novosibirsk, Russia) and the BioMaster LR HS-PCR (2X) mix (Biolabmix, Russia), according to the laboratory protocol for the SMART-9N study [11]. For enzymatic fragmentation and adapter ligation, we used the FTP Display and LIB Display kits (BioLink, Russia, Novosibirsk). cDNA purification was conducted with magnetic beads using VAHTS DNA Clean Beads (Vazyme, Nanjing, China).

To achieve greater HuNoV genome coverage using NGS, two panels of oligonucleotides were designed for the most prevalent genotypes in Russia: GII.4 (Table 1) and GII.7 (Table 2). The synthesized primers covered the norovirus genome fragments within 800 base pairs for the GII.4 genotype and 1000 base pairs for the GII.7 genotype. One-step RT-PCR was performed to amplify the target fragments of the full-length genome, using the Biolabmix-Premium kit (Biolabmix, Russia). Subsequent steps were executed according to the SMART-9N protocol [11]. Amplified fragments were visualized on a 2% agarose gel (TBE buffer, 180 V for 30 min). Libraries prepared following the SMART-9N protocol and those obtained via enrichment methods were pooled into a single tube. The concentration of the final library pool was measured using a Qubit fluorometer (Thermo Fisher Scientific, Eugene, OR, USA) with the QubitTM dsDNA HS Assay Kit (Thermo Fisher Scientific, USA). Genomic libraries were sequenced on the Illumina MiSeq platform, utilizing the MiSeq Reagent Kit v2 for 300 cycles.

### 2.6. Bioinformatic and Phylogenetic Analysis of Nucleotide/Amino Acid Sequences from the ORF1/ORF2 Fragment Within the Norovirus Genome

Bioinformatic analysis of the identified nucleotide sequences from the ORF1/ORF2 fragment in the norovirus genome was conducted according to the algorithm established in our previously published study [12]. Assembly of consensus sequences, generated from forward/reverse read typing results, was performed using UGENE software, version 52 [13]. The number of consensus sequences included in the phylogenetic analysis was 6 for GII.7[P7] and 8 for GII.4[P16], with an average nucleotide fragment length of 490 bp. Genetic distance between amino acid sequences of norovirus genotypes was assessed using the p-distance matrix. We used the Neighbor-Joining method with the Kimura-2 parameter model for the reconstruction of additive phylograms and the calculation of pairwise distances among taxa in the GI and GII genogroups. Nucleotide sequences from the norovirus genome fragment generated in this study are deposited in GenBank, NCBI, with a total of n = 183 sequences under the following accession numbers: PP406492—PP406500; PP486238—PP486248; PQ804494—PQ804497; PP316687—PP316695; PP406485—PP406491; PQ780080.1; PQ780096—PQ780098; PQ803147—PQ803156; PQ804488—PQ804493; PQ596135—PQ596136; PQ596184—PQ596185; PQ614855—PQ614859; PQ614895—PQ614897; PQ614902—PQ614903; PQ780062—PQ780063; PQ780076—PQ780079; PQ596115—PQ596134; PP694327; PQ596096—PQ596114; PV104320; PV104325—PV104336; PP658520—PP694326; PV104293—PV104300; PV104308—PV104319; PV104210—PV104214; PV104278—PV104292; PV104188; PV104191—PV104192.

### 2.7. Bioinformatic and Phylogenetic Analysis of Full-Length Genome Sequences Belonging to Genotypes GII.4 and GII.7

Genome library construction for genotypes GII.4 and GII.7 utilized DRAGEN v. 4.3 algorithms (Sequencing Technology: Illumina). Resulting full-length genome nucleotide sequences were identified using the BLAST service (ver. 2.17.0) [14]. Phylogenetic analysis employed the maximum likelihood method with the parametric Kimura-2 model. Genetic distance between complete amino acid sequences of norovirus genotypes was assessed using a p-distance matrix. Three-dimensional visualization of synonymous and nonsynonymous substitutions in peptide structures was conducted in the SWISS-MODEL client–server application [15]. Nucleotide sequences of the full-length HuNoV genome generated in this study have been deposited in GenBank, NCBI. The total number of deposited sequences amounts to n = 8 under accession numbers PV746275.1-PV746278.1 for GII.4[P16] and PV746280.1, PV806172-PV806174 for GII.7[P7].

## 3. Results

### 3.1. The Genotypic Composition of HuNoV Infection in the Sverdlovsk Region

During the studied period in 2024, the total number of analyzed fecal samples from patients with norovirus gastroenteritis confirmed by ELISA and/or real-time PCR amounted to n = 384. Following Sanger sequencing of the ORF1/ORF2 fragment in the HuNoV genome, n = 220 samples (57.3%) were successfully typed. The highest proportion of identified HuNoV strains corresponds to the second genetic group (GII—92%), while GI was less detected (8%). The detected HuNoV genotypic composition predominantly includes three capsid variants, GII.4, GII.7, and GII.17, along with their corresponding polymerase types, GII.P16, GII.P7, and GII.P17, respectively (Figure 3). The minor genotypic profile comprises capsid variants GII.3, GII.6, GII.2, GIV.1, GI.2, GI.7, and GI.6 and the following polymerase types: GII.P12, GII.P7, GI.P7, and GI.P2.

The HuNoV strains included in the NGS analysis were randomly chosen from the total pool of GII.4[P16] and GII.7[P7] genotypes identified in the Sverdlovsk region during 2024. For this random sample of n = 10, threshold cycle numbers were determined using real-time PCR (Table 3). According to NGS results, eight full-lengths genome sequences corresponding to two of the most prevalent genotypes from the Sverdlovsk region, GII.4[P16] and GII.7[P7], were assembled. The median coverage for the NGS complete-genome nucleotide sequences, using the de novo approach, was 603 for GII.4[P16] and 246 for GII.7[P7].

### 3.2. Phylogenetic Analysis of Nucleotide Sequences from the ORF1/ORF2 Fragment in the HuNoV Genome for the Most Prevalent GII.4 and GII.7 Strains Identified in the Sverdlovsk Region in 2024

When analyzing the additive phylogram with the identified strain GII.4[P16] (Figure 3), a common internal node form indicated close phylogenetic relationships among the sequences. In 2024, the overwhelming majority of nucleotide sequences from the capsid and polymerase type GII.4[P16] underwent no significant evolutionary changes in the RNA polypeptide chain. Despite a relatively genetically homogeneous sample of GII.4[P16] strains from the Sverdlovsk region, some distinct internal clades with the most divergent strains, such as PP406489.1 and PQ804497.1, emerge. The percentage values for genetic distance in these highly heterogeneous strains compared to adjacent GII.4[P16] strains from other municipalities in the Sverdlovsk region ranged from 8% to 12%. Other GII.4[P16] strains that presented in the municipalities exhibited percentage identity values ranging from 96% to 100%.

In the same additive phylogram (Figure 4), the cluster of the Sverdlovsk strain GII.7[P7] forms paraphyletic ties with strains from Nizhny Novgorod (Russian Federation) and Japan and shows the lowest genetic distance of 1% to 3%. The formation of distinct internal nodes is recorded for strains PP486245.1 and PQ596106.1. The matrix for genetic distances in amino acid sequences among the most divergent GII.7[P7] strains shows a 15% distance compared to other 2024 GII.7[P7] genovariants identified in our region.

### 3.3. Phylogenetic Analysis of Nucleotide Sequences from Full-Length Genomes of Norovirus Capsid Variants GII.4 and GII.7

Concerning the reconstruction of HuNoV GII.4 phylogenetic events for nucleotide sequences from full-length genomes, the topology of the cladogram demonstrates the formation of two distinct nodes with clusters of GII.4 strains from the Sverdlovsk region. (Figure 5). One cluster, including strains PV746277.1 and PV746276.1, forms polyphyletic relationships with GII.4 strains from Nizhny Novgorod, the USA, and India. Another cluster, consisting of strains PV746275.1 and PV746278.1, establishes polyphyletic connections with a cluster of strains from the USA and Japan.

Concerning the reconstruction of the additive phylogram, a formation of terminal nodes with varying rates of evolutionary events for clusters of GII.4 strains from the Sverdlovsk region is recorded (Figure 6). When constructing a matrix of genetic distances for amino acid sequences from these GII.4 strains, low levels of divergence were observed, ranging from 1 to 2%. The genetic distance between GII.4 strains from the Sverdlovsk region and strains from other regions in Russia, as well as from other countries, varied between 3% and 5%.

Phylogenetic reconstruction, involving nucleotide sequences from full-length genomes of GII.7[P7] noroviruses, shows the formation of an internal clade with a hypothetical common ancestor among GII.7[P7] strains from the Sverdlovsk region and Nizhny Novgorod (Figure 7). These GII.7[P7] strains, forming a common cluster with sequences from Nizhny Novgorod, also exhibit polyphyletic relationships with sequences from Japan. The genetic distance for amino acid sequences from these strains to current sequences from Nizhny Novgorod and Japan identified during the post-COVID-19 period is 1–2%. The genetic distance for amino acid sequences from strains in the Sverdlovsk region to GII.7[P7] sequences found during the pre-COVID-19 period varies between 16 and 18%. Analysis of the cladogram topology reveals the formation of a distinct internal clade with a terminal node at strain PV746280.1. When constructing a genetic distance matrix, strain PV746280.1 demonstrates the highest genetic distance for amino acid sequences relative to adjacent GII.7[P7] strains from the Sverdlovsk region, with a divergence percentage ranging from 24 to 35%.

### 3.4. Three-Dimensional Models of the Major Capsid Protein VP1 from GII.4 and GII.7 HuNoVs

Three-dimensional modeling of the structural protein VP1 in GII.4 norovirus strains revealed several non-synonymous substitutions at sites within amino acid sequences encoding antigenic determinants responsible for receptor-mediated endocytosis into target cells (Figure 8). In the identified GII.4[P16] strains, amino acid substitutions were found at site H (His) [site 297] => R (Arg) and N (Asn) [site 372] => D (Asp), which are part of epitope A, critical in the conformational integrity of the major capsid protein VP1. Additionally, the identified amino acid substitutions H (His) [site 297] => R (Arg), N (Asn) [site 372] => D (Asp), and V (Val) [site 317] => I (Ile) are located within the region encoding the P2 subdomain of the major capsid protein VP1, essential for viral penetration into target cells. The amino acid substitution at site V (Val) [site 47] => I (Ile) positioned in the domain N/S plays a role in the formation of the VP1 shell. Full-length reference genomes from GenBank used to determine non-synonymous amino acid substitutions in GII.4[P16] strains from the Sverdlovsk region include PQ214835.1, PQ195931.1, OP901694.1, OP712198.1, MW305632.1, MK762568.1, MK756038.1, MG892929.3, LC790068.1, LC790061.1, LC777250.1, LC769714.1, LC769708.1, LC769691.1, LC175468.1, KY947550.1, KY947549.1, KY887604.1, and KY887601.1.

Similarly, the analysis of 3D modeling results for structural protein VP1 from GII.7 norovirus strains shows two identical non-synonymous amino acid substitutions in two GII.7[P7] norovirus strains (Figure 9). The strain PV806172.1 exhibited a substitution at the site encoding the P1 domain in region D of the VP1 structural protein—M (Met) [site 237] => K (Lys); the strain PV806173.1 showed a substitution at the site encoding the VP1 capsid shell—G (Gly) [site 171] => S (Ser). Full-length reference genomes from GenBank used to identify non-synonymous amino acid substitutions in GII.7[P7] strains from the Sverdlovsk region include PQ100947.1; PQ100948.1; LC877021.1; LC877023.1; PQ100946.1; LC877024.1; PQ507186.1; LC877026.1; PQ594187.1; LC877033.1; MW305550.1; MW305606.1; MW305602.1; MZ292759.1; LC122887.1; LC122889.1; MW284778.1; MF140647.1; MT731332.1; and MK762722.1.

## 4. Discussion

This study focused on analyzing the molecular–genotypic composition of HuNoV pathogens identified in the Sverdlovsk region in 2024. It demonstrated that the genetic diversity among noroviruses falls into two genetic groups (GI and GII), and it examined the circulation patterns of identified genovariants along with evolutionary events within various genovariants. HuNoV genotyping results, obtained through Sanger sequencing targeting the ORF1/ORF2 fragments in the norovirus genome, reveal a trend toward an increase in confirmed cases of GII genogroup HuNoV AGE in 2024. This trend aligns with global patterns in genetic group circulation, where noroviruses from the GII genogroup dominated both pediatric and adult populations [16]. The results also demonstrate the emergence of two genotypic profiles: a major genotypic profile and a minor genotypic profile. The major genotypic profile primarily includes GII.4[P16], GII.7[P7], and GII.17[P17]. A significant distinction to the major genotypic profile in 2024, compared to previous analytical periods, is the increasing number of cases linked to the emerging recombinant genovariant GII.7[P7]. This latter has displaced the previously dominant GII.7[P17] [17], which actively circulated from 2022 to 2023 [12]. The higher incidence of GII.4[P16] noroviruses correlates with results from studies conducted in various countries, where GII.4[P16] emerges as the most frequently encountered genovariant in HuNoV gastroenteritis cases [5,12,18,19]. Conversely, reports regarding the widespread presence of recombinant GII.7[P7] remain absent, except in certain regions in Russia and Japan, suggesting the unique nature of the major HuNoV landscape identified in the Sverdlovsk region in 2024. Over the studied period, other registered HuNoV strains from genogroups GI and GII included types GII.3[P12], GII.6[P7], GII.2[Px], GIV.1[Px], GI.6[P11], GI.7[P7], and GI.2[P2]. The 2024 genotype distribution of HuNoVs in the minor genotypic profile further confirms the uniqueness of the recorded genotypic structure in the Sverdlovsk region for the analytical period.

Phylogenetic analysis results for the ORF1/ORF2 fragment from the HuNoV genome reveal paraphyletic connections among strains originating from Japan, Brazil, the USA, Spain, and other countries for the two most common genotypes GII.4[P16] and GII.7[P7]. The increased number of amino acid sequences for GII.4[P16] found in this study demonstrates minimal genetic distance from non-Russian strains, ranging from 98% to 100% identity, indicating the widespread dissemination of GII.4[P16] genotypes. In interpreting phylogenetic events related to GII.7[P7], clustering occurs with sequences from Japan and Nizhny Novgorod, exhibiting an identity degree of 98% to 100%. The high percentage of homology within the analyzed segment of the norovirus genome indirectly might suggest the potential importation of recombinant GII.7[P7] from Japan into Russian territory.

Phylogenetic analysis of HuNoV full-length genomes GII.4[P16] and GII.7[P7] shows similar results to those identified in phylogenetic analysis of ORF1/OFR2 sequences. HuNoV full-length phylogenetic analysis for GII.4[P16] indicates the formation of the smallest genetic distance between strains from the Sverdlovsk region and non-Russian GII.4[P16] strains, showing 95% to 97% identity. A more detailed examination of amino acid sequences from GII.4[P16] strains in the Sverdlovsk region revealed several mutational substitutions, possibly driven by forces of divergent evolution and geographical features influencing circulation. A high percentage of identity between Sverdlovsk strains and non-Russian strains supports evidence of the GII.4[P16] genovariant’s widespread circulation. Phylogenetic analysis results for GII.7[P7] indicate a high degree of homology with strains from Nizhny Novgorod and Japan, with the identity percentage reaching 98–99%. It is important to note that the smallest genetic distance with Sverdlovsk strains is observed in non-Russian strains identified during the post-COVID-19 period from 2022 to 2024. A comparison of genetic distances in amino acid sequences between Sverdlovsk strains and non-Russian strains identified in the pre-COVID-19 period shows an increase in divergence percentages, reaching 16–18%. The strain GII.7[P7] PV746280.1 forms a distinct internal clade, whereas divergence among other Sverdlovsk strains varies between 24 and 35%. The largest genetic distance between Sverdlovsk GII.7[P7] strains and strains originating from other countries developed during the studied evolutionary events may result from driving forces behind divergent evolution through reproductive isolation processes following multiple passages of recombinant GII.7[P7] in the human population [20,21].

Analysis of results from 3D models examining nonsynonymous amino acid substitutions in key antigenic determinants of the main capsid protein VP1 revealed several mutational changes in genovariants GII.4[P16] and GII.7[P7]. In the sequenced Sverdlovsk GII.4[P16] strains, approximately four significant mutations appeared that do not occur in other non-Russian strains, including some Russian strains identified in 2022. A greater number of mutational changes in Sverdlovsk GII.4[P16] strains impact amino acid sites in region D of the VP1 capsid protein, responsible for receptor-mediated endocytosis into target cells [22,23]. Closer monitoring of evolutionary events in region D of the VP1 capsid protein will enable predictions regarding the emergence of group illness outbreaks within the human population, particularly where collective immunity is lacking [24]. The identified amino acid substitutions are of critical importance due to their role in evading the adaptive immune response in the human body [23]. The further evolution of established antigenic patterns in binding subdomains will inevitably lead to an increased incidence of HuNoV gastroenteritis. In GII.7[P7] strains from the Sverdlovsk region, only two mutational changes affected amino acid sites responsible for forming the structural protein VP1 shell and the P1 subdomain, both essential for the penetration process into target cells. The high identity of polypeptides among Sverdlovsk GII.7[P7] strains and those from Japan, combined with the absence of significant mutational changes in GII.7[P7] strains in this study, suggests a possible introduction of recombinant GII.7[P7] from Japan into Russian territory, as previously mentioned in ORF1/ORF2 fragment phylogenetic analysis.

One of the main limitations in HuNoV genotyping is the choice of method. Different laboratories may use varying methods and protocols, leading to significant differences in results and reducing the overall reliability and comparability of genotyping data. The main limitation in our study is the number of norovirus strains included in the NGS analysis. We chose five GII.4 strains and five GII.7 strains, as the most prevalent and representative HuNoV genotypes in our region. We could not include more HuNoV strains due to financial limitations and the need to achieve better genome coverage using NGS. For the same reasons, we did not include the less prevalent circulating genotypes despite the vast diversity of HuNoV genotypes in our results.

## 5. Conclusions

Thus, as a result of the molecular characterization and the epidemiology of HuNoV circulating in the Sverdlovsk region, relevant data emerged regarding the genotypic composition of HuNoVs, circulation characteristics, and evolutionary events involving the most common genovariants GII.4[P16] and GII.7[P7]. The presented genotyping system, based on the analysis of the ORF1/ORF2 fragment in the norovirus genome, proves most suitable for routine molecular–genetic monitoring, aiming for continuous updates to the norovirus landscape in the Sverdlovsk region. Studying mutational changes in key antigenic determinants defining interaction specificity with the human host will allow predictions regarding evolutionary developments, to control the epidemiological situation within surveillance systems and to mitigate the impact of emerging HuNoV strains.

## Figures and Tables

**Figure 1 viruses-17-01243-f001:**
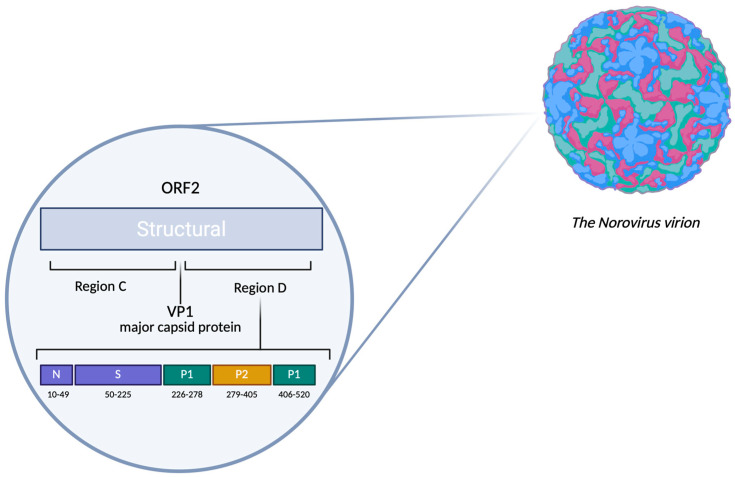
Illustration of the main capsid protein structure—VP1 in the norovirus virion. (The image was created using BioRender web application).

**Figure 2 viruses-17-01243-f002:**
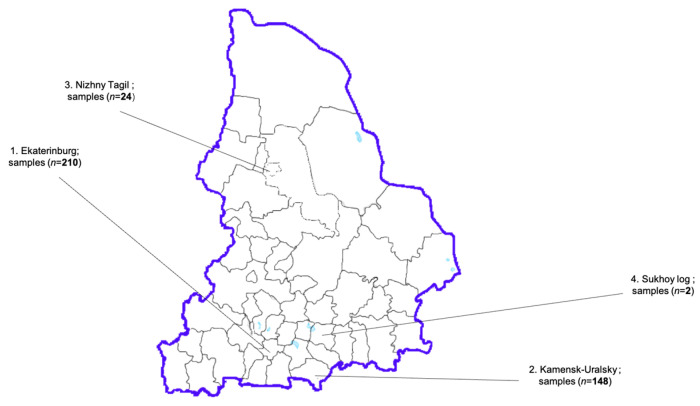
Municipalities of the Sverdlovsk region participating in this study (1—Ekaterinburg, 2—Kamensk-Uralsky, 3—Nizhny Tagil, 4—Sukhoy Log).

**Figure 3 viruses-17-01243-f003:**
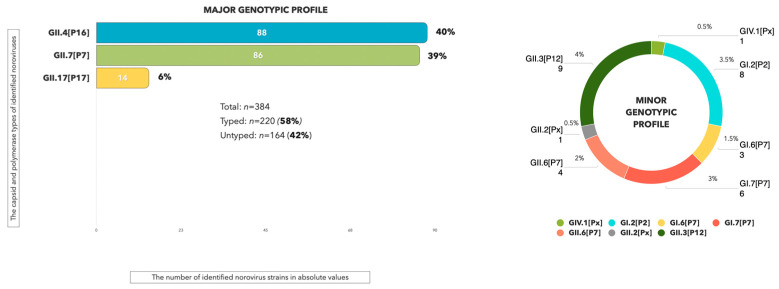
Genotypic composition of HuNoVs identified in the Sverdlovsk region during the analytical period of 2024 (GII.Px—unidentified polymerase types).

**Figure 4 viruses-17-01243-f004:**
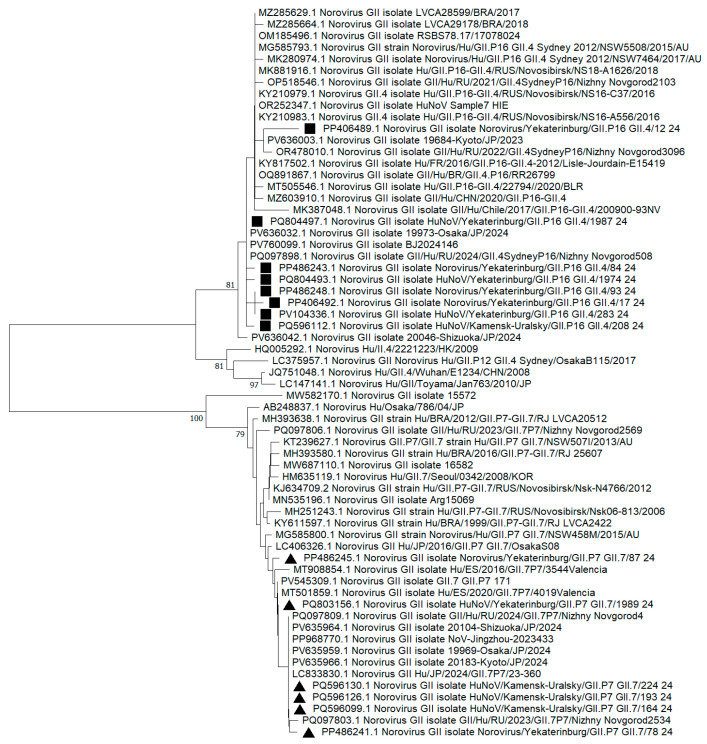
Reconstruction of phylogenetic events for nucleotide sequences from the VP1/RdRp fragment of genovariants GII.4[P16] and GII.7[P7] in 2024, using the Kimura 2-parameter model. Black squares represent genovariant GII.4[P16], and black triangles represent genovariant GII.7[P7], identified in the Sverdlovsk region in 2024.

**Figure 5 viruses-17-01243-f005:**
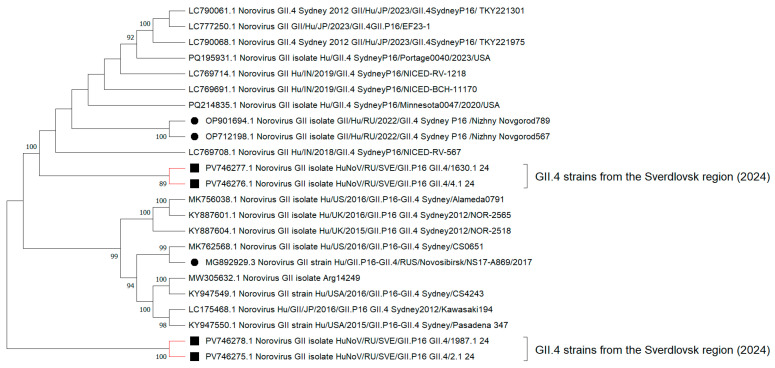
Cladogram of nucleotide sequences from full-length genomes of GII.4 noroviruses (the black square indicates full-length genome sequences of GII.4 norovirus strains identified in the Sverdlovsk region in 2024; the black circle represents full-length genome sequences of GII.4 norovirus strains identified in other regions of Russia over various analytical periods).

**Figure 6 viruses-17-01243-f006:**
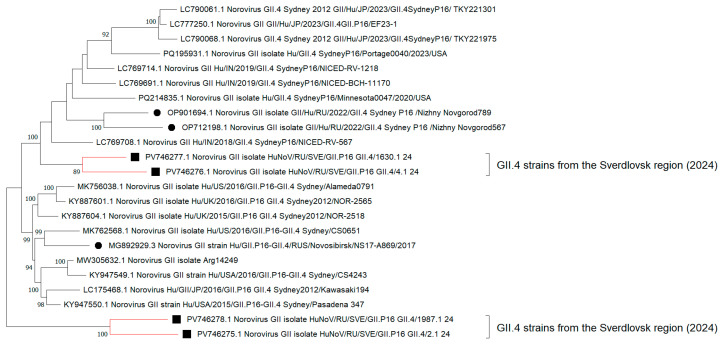
Additive phylogram of nucleotide sequences from full-length genomes of GII.4 noroviruses (the black square indicates full-length genome sequences of GII.4 norovirus strains identified in the Sverdlovsk region in 2024; the black circle represents full-length genome sequences of GII.4 norovirus strains identified in other regions of Russia over various analytical periods).

**Figure 7 viruses-17-01243-f007:**
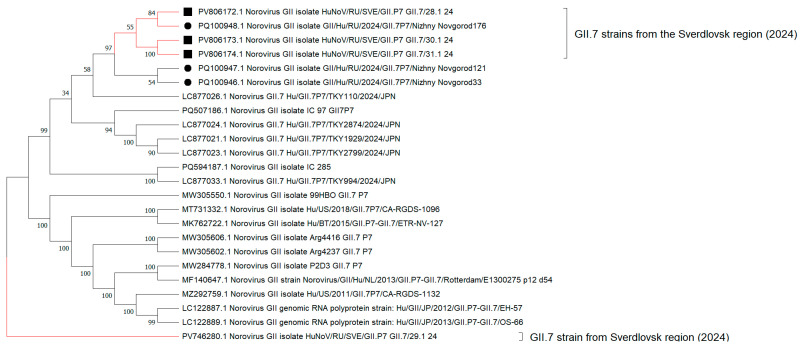
Cladogram of nucleotide sequences from full-length genomes for GII.7 noroviruses (black square indicates full-length genome sequences for GII.7 norovirus strains identified in the Sverdlovsk region in 2024; black circle indicates full-length genome sequences for GII.7 norovirus strains identified in other regions of Russia during 2024).

**Figure 8 viruses-17-01243-f008:**
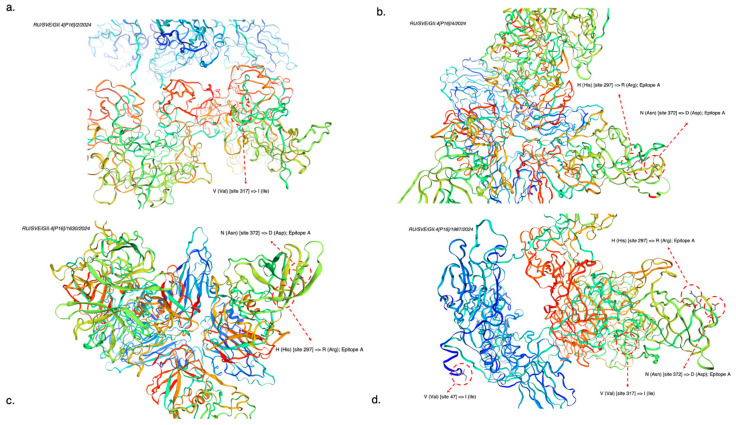
Reconstruction of 3D models showing non-synonymous substitutions in amino acid sequences for the main structural protein VP1 from GII.4 norovirus strains in the Sverdlovsk region. (**a**) Non-synonymous substitutions in VP1 protein from strain RU/SVE/GII.4[P16]/2/2024; (**b**) non-synonymous substitutions in VP1 protein from strain RU/SVE/GII4[P16]/4/2024; (**c**) non-synonymous substitutions in VP1 protein from strain RU/SVE/GII.4[P16]/1630/2024; (**d**) non-synonymous substitutions in VP1 protein from strain RU/SVE/GII.4[P16]/1987/2024) (3D visualization illustrating synonymous/non-synonymous substitutions in peptide structures was created using the SWISS-MODEL client–server application).

**Figure 9 viruses-17-01243-f009:**
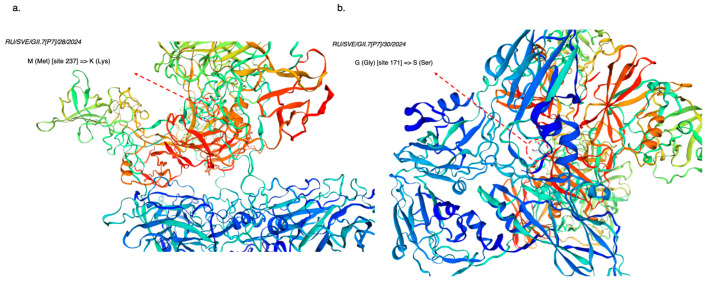
Reconstruction of 3D models showing non-synonymous substitutions in amino acid sequences for the main structural protein VP1 from GII.7 norovirus strains in the Sverdlovsk region. (**a**) Non-synonymous substitutions in the VP1 protein from strain RU/SVE/GII.7[P7]/28/2024; (**b**) non-synonymous substitutions in the VP1 protein from strain RU/SVE/GII.7[P7]/30/2024) (3D visualization illustrating synonymous and non-synonymous substitutions in peptide structures utilized the SWISS-MODEL client–server application).

**Table 1 viruses-17-01243-t001:** Oligonucleotide panel for GII.4 genotyping (average fragment length, 800 bp).

Name	Sequence	Primer Region	Tm Hybridization
NV_GII.4_1F	GTGAATGAAGATGGCGTCTAACGA	1–24	56
NV_GII.4_1R	GAGGTCTTTTATGGGYCTGGTG	807–830	56
NV_GII.4_2F	GCTGCTGGGTGTTAGACCTCA	635–656	56
NV_GII.4_2R	CCTGGCYGCTGCTATTCGAG	1406–1426	56
NV_GII.4_3F	CCACATGACAACCCTGTTGAAAGA	1266–1290	56
NV_GII.4_3R	CACGGGCAATCAGAGAACCG	2027–2047	56
NV_GII.4_4F	GGTCAGCCAGATATGTGGAAGG	1893–1915	56
NV_GII.4_4R	GCTYAAGCCCTTGCTGGAGAATG	2668–2691	56
NV_GII.4_5F	GCTCTGGTCGAAGCCACAATYAG	2513–2536	56
NV_GII.4_5R	GTTCCCATTCTGGCTGCAAGAG	3307–3329	56
NV_GII.4_6F	GCCGRTTCAGRTTCCCAARACCAATT	3195–3220	56
NV_GII.4_6R	GAYGAGATGAAGGCACACTGCATA	4181–4204	56
NV_GII.4_7F	GGCAARCCTCCAAGACCAAGTG	3786–3808	56
NV_GII.4_7R	GGAATTTGCTTGTATAATGTCAGGGG	4546–4572	56
NV_GII.4_8F	AGTCGCTGAGGATCTRCTGG	4393–4412	56
NV_GII.4_8R	CTATTGCGGCACCTGTAGCG	5191–5210	56
NV_GII.4_9F	CTGACTTGAGCACGTGGGAG	5034–5053	56
NV_GII.4_9R	CCCCAGCAGTGCMTTTGTTG	5840–5859	56
NV_GII.4_10F	GATGATGTTTTCACAGTYTCGTGCC	5661–5685	56
NV_GII.4_10R	GGGAGCAGACAGTCCAAATCC	6425–6445	56
NV_GII.4_11F	GCTGTAGCCCCCACCTTTC	6337–6356	56
NV_GII.4_11R	TAGCTCTTCCTGGCAGTGCC	7112–7132	56
NV_GII.4_12F	CAAATTGAGGCCACYAAAAAGCTAC	6095–6929	56
NV_GII.4_12R	TGGACTGGCGCTTTYAACACG	7439–7459	56

**Table 2 viruses-17-01243-t002:** Oligonucleotide panel for GII.7 genotyping (average fragment length, 1000 bp).

Name	Sequence	Primer Region	Tm Hybridization
NV_GII.7_1F	GTGAATGAAGATGGCGTCTAACGA	1–24	56
NV_GII.7_1R	GCAAYTCAAAATCACCTATCAGGG	980–1004	56
NV_GII.7_2F	GGACGTTTGCAGGYATAGTRGAG	888–910	56
NV_GII.7_2R	TTAAAATGGTCYTTCCACATGTCRG	1914–1939	56
NV_GII.7_3F	GAGGCGTGTTCTAGGAGAGTTG	1828–1850	56
NV_GII.7_3R	CTTGGCYTCTTCCTCTTCACAG	2823–2845	56
NV_GII.7_4F	GAGTTCAAGAGAATYAGGGAAGAAAG	2708–2733	56
NV_GII.7_4R	TYATGAGAGAYCAGTTGAGRCCCTT	3750–3774	56
NV_GII.7_5F	CACCTACTGTGGWGCCCCAAT	3582–3603	56
NV_GII.7_5R	ARGAGTGAGCTTGGACWACATCTG	4550–4574	56
NV_GII.7_6F	GTCATCTCAGTCCAGGARGG	4435–4455	56
NV_GII.7_6R	AGCTGTGAACGCGTTCCCAG	5406–5426	56
NV_GII.7_7F	CAAGCTCCTGCAGGTGAGTTY	5250–5270	56
NV_GII.7_7R	ACTGGTGGRGACCARTAYGCC	6150–6170	56
NV_GII.7_8F	CTACCAGAGCCCATGAAGCYAAC	6120–6143	56
NV_GII.7_8R	CATATGAAGCAGACTGCAYAGG	7072–7094	56
NV_GII.7_9F	ATGCCGCRAGGGGYTCTGTY	6951–6971	56
NV_GII.7_9.1R	CTCTTCGCCCAYYTSCGTAR	7433–7452	56
NV_GII.7_9.2R	AGRTTTAGTGAAAAGATYRRTTAGGAAAG	7467–7495	56

**Table 3 viruses-17-01243-t003:** Results of cycle threshold (Ct) determination using real-time PCR (reagent kit—AmpliSens^®^ Norovirus GI-GII-FL, Central Scientific Research Institute of Epidemiology, Moscow, Russian Federation) (Target IPC—Internal Positive Control; Target GII—Genogroup GII).

Sample	Ct IPC (Mean)	Ct GII (Mean)	Med. Coverage Full-Length Sequence	AC Numbers, NCBI GenBank
2/GII.4[P16]	28.97	14.42	1621	PV746275.1
4/GII.4[P16]	29.26	19.27	223	PV746276.1
1629/GII.4[P16]	29.48	27.67	1	—
1630/GII.4[P16]	29.95	22.06	162	PV746277.1
1987/GII.4[P16]	29.33	21.42	406	PV746278.1
28/GII.7[P7]	30.57	23.47	100	PV806172.1
29/GII.7[P7]	29.75	14.13	900	PV746280.1
30/GII.7[P7]	30.17	19.55	30	PV806173.1
31/GII.7[P7]	30.91	19.87	44	PV806174.1
38/GII.7[P7]	34.13	32.05	1	—

## Data Availability

The raw data supporting the conclusions of this article will be made available by the authors on request.

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
