# Peer review of "Molecular Characterization and Epidemiology of Human Noroviruses in the Sverdlovsk Region, Russian Federation"

_viruses, 2025, doi:10.3390/v17091243_

Round 1

Reviewer 1 Report

Comments and Suggestions for Authors

The authors of this manuscript analyzed 384 patient samples and characterized 220 norovirus strains in the Sverdlovsk region in 2024. The most prevalent genotypes were GII.4 (40%), GII.7 (39%), and GII.17 (6%). Using NGS, researchers assembled eight complete genomes of the dominant GII.4[P16] and GII.7[P7] strains, discovering non-synonymous substitutions that suggest these viruses are evolving to potentially evade immune responses. This research provides crucial insights into the current genotypic composition and evolutionary dynamics of circulating noroviruses. underscoring the importance of continued genomic surveillance to predict future outbreaks. The manuscript is well written. The manuscript is well-written. However, the information in Figure 4 is also represented in Figure 5; therefore, please remove Figure 4.

Author Response

Thank you very much for taking the time to review this manuscript. Please find the detailed responses below and the corresponding revisions.

Concerning our reply to Reviewer 1 comments, we acknowledge that figures 4 and 5 may look similar, but the significance of each tree is different. Both phylogenetic trees are complementary and underscore the phyletic relationships and the evolutionary distance of GII.4 strains.

Comments from the Editors and Reviewers:

Reviewer 1: The authors of this manuscript analyzed 384 patient samples and characterized 220 norovirus strains in the Sverdlovsk region in 2024. The most prevalent genotypes were GII.4 (40%), GII.7 (39%), and GII.17 (6%). Using NGS, researchers assembled eight complete genomes of the dominant GII.4[P16] and GII.7[P7] strains, discovering non-synonymous substitutions that suggest these viruses are evolving to potentially evade immune responses. This research provides crucial insights into the current genotypic composition and evolutionary dynamics of circulating noroviruses. underscoring the importance of continued genomic surveillance to predict future outbreaks. The manuscript is well written.

Comments 1: However, the information in Figure 4 is also represented in Figure 5; therefore, please remove Figure 4.

Response:

We thank the reviewer for this point. Two figures are presented in our paper and we prefer to retain them both. Figure 4 shows a cladogram depicting the phyletic relationships among the GII.4 taxa; this figure is necessary to represent the overall structure of the phylogram topology. However, Figure 5 shows an additive phylogram in which the length of each terminal branch represents the evolutionary distance between taxa. To preserve the overall structure and to keep the results of our publication informative, it is important to keep both figures in the manuscript. Figures 4 and 5 confirm what is written in the manuscript about the phyletic relationships and the evolutionary distance of GII.4 strains (lines 263-265; 270-276).

On behalf of all the authors, thank you for your kind consideration!

Roman O. Bykov, Postgraduate student of the Federal Budgetary Institution of Science State Scientific Center of Virology and Biotechnology "Vector" Junior Research Assistant in the Laboratory of Enteric Viral Infections. Ekaterinburg Research Institute of Viral Infections, Federal Budgetary Institution of Science «Federal Scientific Research Institute of Viral Infections «Virome» Federal Service for Surveillance on Consumer Rights Protection and Human Wellbeing Rospotrebnadzor, 620030, Ekaterinburg, Russian Federation +7(982) 690-69-64

Reviewer 2 Report

Comments and Suggestions for Authors

General overview:

The study conducted by the authors addresses an important public health topic, as human noroviruses (HuNoVs) are recognized as a leading cause of acute gastroenteritis worldwide. The work focuses on the genotypic composition of HuNoVs circulating in China during 2024, employing both Sanger sequencing and next-generation sequencing. This study makes contributions into the molecular epidemiology of HuNoVs and supports a better understanding of their role in acute diarrheal disease.

The manuscript is concise and generally well-written, although some modifications could be made to enhance its quality and comprehensibility

  • Introduction:

Page 3, Lines 81–83: Figure 1 seems unnecessary, as the information is already clear in the text; if removed, please renumber the remaining figures accordingly.

  • Material and Methods

Page 3, Lines 103-110: The sampling description requires further clarification. Please specify: 

  • the number of municipalities in Sverdlovsk region from which patients were recruited;
  • the total number of fecal samples collected and analyzed;
  • whether the initial laboratory diagnosis was performed locally, and only the positive samples for NoV (n=384) forwarded to Federal Scientific Research Institute of Vital Infections to molecular characterization;
  • or whether all samples were analyzed directly at Institute Federal Scientific Research Institute of Vital Infections.

These clarifications are important to better understand the representativeness of the sampling and the diagnostic workflow.

  • Results

Please consider adding a table summarizing the epidemiological data of the sequenced norovirus cases (age, sex, severity of clinical manifestations when available, and genotype) to better illustrate the clinical–epidemiological context of your phylogenetic findings.

Page 8, Figure 2

In the figure, the bar for genotype GII.7\[P7] should indicate 86 samples, not 82. Please, correct this discrepancy.

Page 8, Lines 234-240     

The authors describe highly heterogeneous genetic divergence among municipalities, but the specific municipalities are not identified. It would improve clarity if the authors specify which municipalities were included in this analysis and, if considered appropriate; add a map of the region highlighting these municipalities.

Page 9, Figure 3

Please revise the legend of Figure 3 for clarity. A suggested wording is: “Reconstruction of phylogenetic events for nucleotide sequences from the VP1/RdRp fragment of genovariants GII.4\[P16] and GII.7\[P7] in 2024, using the Kimura 2-parameter model. Black squares represent genovariants GII.4\[P16], and black triangles represent genovariants GII.7\[P7] identified in the  Sverdlovsk region in 2024”.

  • Discussion

Page 14, Lines 402-407

Consider adding a brief note on the patients’ clinical course and exploring whether the observed amino acid substitutions (e.g., in GII.4[P16]) could be related to clinical manifestations. If data on disease severity are available, incorporating this information could further highlight the potential clinical relevance of the genetic findings.

The manuscript would be strengthened by the inclusion of a brief discussion of the study’s limitations.

Author Response

Thank you very much for taking the time to review this manuscript. Please find the detailed responses below and the corresponding revisions.

Concerning the comments from Reviewer 2, we have addressed each separately. Please note that the table summarizing the epidemiological data proposed by reviewer 2 in point 3, has been added as a supplementary file. The table is very large to be published in the main manuscript. A new figure was added (figure 2) showcasing all the included municipalities in this study.

Comments from the Editors and Reviewers:

Reviewer 2: The study conducted by the authors addresses an important public health topic, as human noroviruses (HuNoVs) are recognized as a leading cause of acute gastroenteritis worldwide. The work focuses on the genotypic composition of HuNoVs circulating in China during 2024, employing both Sanger sequencing and next-generation sequencing. This study makes contributions into the molecular epidemiology of HuNoVs and supports a better understanding of their role in acute diarrheal disease.

The manuscript is concise and generally well-written, although some modifications could be made to enhance its quality and comprehensibility.

Comments 1: Page 3, Lines 81–83: Figure 1 seems unnecessary, as the information is already clear in the text; if removed, please renumber the remaining figures accordingly.

Response:

We thank the reviewer for this point. Figure 1 depicts a simplified representation of the structure of the major capsid protein VP1 in noroviruses, including the main antigenic determinants responsible for mediating norovirus penetration into the target cell and eliciting specific immune responses in humans. We consider it reasonable to retain Figure 1 in the manuscript to visually demonstrate the role of norovirus structural proteins in the infection process. The structural proteins and their components are cited in the introduction and the discussion. Therefore, this would allow readers that are specialists in norovirus research or scientists encountering this topic for the first time to better understand this norovirus topic.

Comments 2: Material and Methods. Page 3, Lines 103-110: The sampling description requires further clarification. Please specify:  the number of municipalities in Sverdlovsk region from which patients were recruited; the total number of fecal samples collected and analyzed; whether the initial laboratory diagnosis was performed locally, and only the positive samples for NoV (n=384) forwarded to Federal Scientific Research Institute of Vital Infections to molecular characterization; or whether all samples were analyzed directly at Institute Federal Scientific Research Institute of Vital Infections. These clarifications are important to better understand the representativeness of the sampling and the diagnostic workflow.

Response:

We thank the reviewer for this important comment. In 2024, the collection of stool samples from patients with a confirmed diagnosis of HuNoVs was organized in four municipalities: Ekaterinburg, Kamensk-Uralsky, Nizhny Tagil, and Sukhoy Log. The total number of “positive” biological samples collected during the 2024 analytical period was n=384. Samples submitted to our institute were screened by either an enzyme immunoassay or qPCR at two medical facilities: The Ekaterinburg Consultative Diagnostic Center and the State Autonomous Healthcare Institution the Sverdlovsk Region Children’s City Hospital (Kamensk-Uralsky). Figure 2 was added as proposed in comment 5.

This was added to the manuscript (lines 106-116), and the text now read:

«The collection of stool samples from patients with a confirmed diagnosis of HuNoVs was organized in four municipalities during 2024: Ekaterinburg, Kamensk-Uralsky, Nizhny Tagil, and Sukhoy Log. Samples from patients were screened for HuNoVs by either an enzyme immunoassay (Norovirus-antigen-enzyme immunoassays-Best, Vector-Best, Novosibirsk, Russian Federation) or qPCR (AmpliSens® Norovirus GI-GII-FL, Central Scientific Research Institute of Epidemiology, Moscow, Russian Federation) at the following facilities: The Ekaterinburg Consultative Diagnostic Center and the State Autonomous Healthcare Institution the Sverdlovsk Region Children’s City Hospital (Kamensk-Uralsky). Only positive samples were submitted to the Federal Scientific Research Institute of Viral Infections «Virome» (n=384) for further molecular analysis»

Comments 3: Please consider adding a table summarizing the epidemiological data of the sequenced norovirus cases (age, sex, severity of clinical manifestations when available, and genotype) to better illustrate the clinical–epidemiological context of your phylogenetic findings.

Response: We have revised the table in accordance with the comments provided. Due to the large volume of data, it was decided to add the table to the supplementary materials section.

Table in the supplementary materials section. Data on identified cases of norovirus infection. (sample number, sex, age and genotype)

Number

Sex

Age

Genotype

2

-

2

GII.4[P16]

4

-

5

GII.4[P16]

9

-

5

GII.4[P16]

10

-

3

GII.4[P16]

11

-

5

GII.4[P16]

13

-

4

GII.4[P16]

14

-

3

GII.4[P16]

15

-

3

GII.4[P16]

16

-

2

GII.4[P16]

27

-

3

GII.4[P16]

61

-

5

GII.4[P16]

63

male

13

GII.4[P16]

79

-

3

GII.4[P16]

83

-

4

GII.4[P16]

84

-

9

GII.4[P16]

92

-

3

GII.4[P16]

93

-

3

GII.4[P16]

102

male

3

GII.4[P16]

105

female

5

GII.4[P16]

107

female

12

GII.4[P16]

108

female

13

GII.4[P16]

110

female

5

GII.4[P16]

111

male

4

GII.4[P16]

154

female

-

GII.4[P16]

155

female

-

GII.4[P16]

159

female

8

GII.4[P16]

163

male

3

GII.4[P16]

190

male

-

GII.4[P16]

192

male

7

GII.4[P16]

201

female

3

GII.4[P16]

208

female

12

GII.4[P16]

210

female

-

GII.4[P16]

231

male

6

GII.4[P16]

251

male

6

GII.4[P16]

253

male

3

GII.4[P16]

254

female

-

GII.4[P16]

259

female

15

GII.4[P16]

383

-

-

GII.4[P16]

384

-

-

GII.4[P16]

390

-

-

GII.4[P16]

400

female

23

GII.4[P16]

401

female

8

GII.4[P16]

402

female

3

GII.4[P16]

403

male

8

GII.4[P16]

404

female

-

GII.4[P16]

405

male

13

GII.4[P16]

456

male

4

GII.4[P16]

457

male

3

GII.4[P16]

459

-

11

GII.4[P16]

500

-

3

GII.4[P16]

501

-

4

GII.4[P16]

508

-

5

GII.4[P16]

511

-

16

GII.4[P16]

513

-

2

GII.4[P16]

514

-

2

GII.4[P16]

602

-

2

GII.4[P16]

604

-

5

GII.4[P16]

606

-

3

GII.4[P16]

608

-

2

GII.4[P16]

609

-

3

GII.4[P16]

612

-

12

GII.4[P16]

614

-

1

GII.4[P16]

619

-

2

GII.4[P16]

680

female

4

GII.4[P16]

687

male

6

GII.4[P16]

690

male

4

GII.4[P16]

698

male

3

GII.4[P16]

707

male

15

GII.4[P16]

708

female

2

GII.4[P16]

819

-

6

GII.4[P16]

1001

-

5

GII.4[P16]

1107

male

6

GII.4[P16]

1213

female

6

GII.4[P16]

1264

-

3

GII.4[P16]

1266

-

3

GII.4[P16]

1272

-

3

GII.4[P16]

1273

-

4

GII.4[P16]

1275

-

2

GII.4[P16]

1276

-

2

GII.4[P16]

1629

female

3

GII.4[P16]

1630

female

3

GII.4[P16]

1633

female

3

GII.4[P16]

1638

male

4

GII.4[P16]

1640

female

4

GII.4[P16]

1679

-

4

GII.4[P16]

1680

-

2

GII.4[P16]

1850

-

4

GII.4[P16]

1987

-

6

GII.4[P16]

1

-

3

GII.7[P7]

8

-

2

GII.7[P7]

26

-

3

GII.7[P7]

28

male

7

GII.7[P7]

29

male

3

GII.7[P7]

30

female

9

GII.7[P7]

31

female

11

GII.7[P7]

34

female

11

GII.7[P7]

36

female

9

GII.7[P7]

37

male

9

GII.7[P7]

38

female

-

GII.7[P7]

39

female

-

GII.7[P7]

40

female

-

GII.7[P7]

41

female

-

GII.7[P7]

42

male

21

GII.7[P7]

43

male

8

GII.7[P7]

44

female

10

GII.7[P7]

45

female

4

GII.7[P7]

46

female

5

GII.7[P7]

64

-

16

GII.7[P7]

65

-

19

GII.7[P7]

66

-

6

GII.7[P7]

67

-

12

GII.7[P7]

68

male

5

GII.7[P7]

78

-

4

GII.7[P7]

85

-

9

GII.7[P7]

87

-

15

GII.7[P7]

88

-

12

GII.7[P7]

90

-

12

GII.7[P7]

106

female

3

GII.7[P7]

156

female

4

GII.7[P7]

157

female

3

GII.7[P7]

158

male

10

GII.7[P7]

161

female

11

GII.7[P7]

162

-

3

GII.7[P7]

164

female

10

GII.7[P7]

165

male

17

GII.7[P7]

166

male

4

GII.7[P7]

168

female

3

GII.7[P7]

171

female

14

GII.7[P7]

175

female

8

GII.7[P7]

177

male

2

GII.7[P7]

178

female

9

GII.7[P7]

179

female

3

GII.7[P7]

180

male

7

GII.7[P7]

181

female

4

GII.7[P7]

186

male

3

GII.7[P7]

191

male

4

GII.7[P7]

200

-

2

GII.7[P7]

202

female

11

GII.7[P7]

211

male

3

GII.7[P7]

215

female

5

GII.7[P7]

218

female

6

GII.7[P7]

219

female

6

GII.7[P7]

220

female

2

GII.7[P7]

222

female

10

GII.7[P7]

223

-

4

GII.7[P7]

224

male

7

GII.7[P7]

228

female

8

GII.7[P7]

233

-

3

GII.7[P7]

252

male.

3

GII.7[P7]

256

male

18

GII.7[P7]

258

male

9

GII.7[P7]

261

-

-

GII.7[P7]

263

-

6

GII.7[P7]

267

-

21

GII.7[P7]

268

male

9

GII.7[P7]

269

male

13

GII.7[P7]

271

female

8

GII.7[P7]

274

female

21

GII.7[P7]

276

male

5

GII.7[P7]

277

female

5

GII.7[P7]

278

male

4

GII.7[P7]

279

female

3

GII.7[P7]

283

male

4

GII.7[P7]

284

-

4

GII.7[P7]

453

male

17

GII.7[P7]

454

male

7

GII.7[P7]

460

male

12

GII.7[P7]

461

male

16

GII.7[P7]

462

female

10

GII.7[P7]

464

male

10

GII.7[P7]

610

-

3

GII.7[P7]

620

female

14

GII.7[P7]

622

female

1

GII.7[P7]

627

male

7

GII.7[P7]

628

-

3

GII.7[P7]

629

-

2

GII.7[P7]

25

-

5

GII.17[P17]

387

-

-

GII.17[P17]

388

male

-

GII.17[P17]

389

-

-

GII.17[P17]

475

-

-

GII.17[P17]

611

-

12

GII.17[P17]

659

-

1

GII.17[P17]

823

female

12

GII.17[P17]

824

female

9

GII.17[P17]

825

-

15

GII.17[P17]

827

-

14

GII.17[P17]

1220

female

1

GII.17[P17]

1223

-

13

GII.17[P17]

1692

-

2

GII.17[P17]

59

female

2

GIV.1

234

-

17

GI.2[P2]

241

-

12

GI.2[P2]

250

female

10

GI.2[P2]

1860

-

3

GI.2[P2]

1873

female

2

GI.2[P2]

1877

male

2

GI.2[P2]

1881

female

5

GI.2[P2]

1761

female

3

GI.2[P2]

1762

-

9

GI.7[P7]

1763

male

10

GI.7[P7]

1766

-

11

GI.7[P7]

1768

female

8

GI.7[P7]

1769

female

13

GI.7[P7]

70

-

2

GII.6[P7]

71

male

1

GII.6[P7]

184

female

6

GII.6[P7]

199

-

3

GII.6[P7]

72

-

-

GII.2[Px]

75

-

2

GII.3[P12]

80

-

5

GII.3[P12]

422

-

19

GII.3[P12]

423

female

19

GII.3[P12]

477

-

-

GII.3[P12]

479

-

-

GII.3[P12]

820

-

18

GII.3[P12]

73

-

2

GII.3[P12]

172

male

-

GI.6[P7]

174

-

-

GI.6[P7]

176

-

-

GI.6[P7]

Comments 4: Page 8, Figure 2. In the figure, the bar for genotype GII.7\[P7] should indicate 86 samples, not 82. Please, correct this discrepancy

Response:

We thank the reviewer for his careful reading of our manuscript. We made changes to Figure 8.

Comments 5: Page 8, Lines 234-240. The authors describe highly heterogeneous genetic divergence among municipalities, but the specific municipalities are not identified. It would improve clarity if the authors specify which municipalities were included in this analysis and, if considered appropriate; add a map of the region highlighting these municipalities.

Response:

Thank you for the suggestion. A figure (Figure 2) showing the main municipalities that participated in the study, has been added to the main text of the manuscript.

Figure 2. Municipalities of the Sverdlovsk region participating in this study (1—Ekaterinburg, 2— Kamensk-Uralsky, 3— Nizhny Tagil, 4—Sukhoy Log). 

Comments 6: Page 9, Figure 3. Please revise the legend of Figure 3 for clarity. A suggested wording is: “Reconstruction of phylogenetic events for nucleotide sequences from the VP1/RdRp fragment of genovariants GII.4\[P16] and GII.7\[P7] in 2024, using the Kimura 2-parameter model. Black squares represent genovariants GII.4\[P16], and black triangles represent genovariants GII.7\[P7] identified in the Sverdlovsk region in 2024”.

Response:

Thank you for this valuable comment. The caption for Figure 3 was revised and now reads:

«Reconstruction of phylogenetic events for nucleotide sequences from the VP1/RdRp fragment of genovariants GII.4[P16] and GII.7[P7] in 2024, using the Kimura 2-parameter model. Black squares represent genovariants GII.4[P16], and black triangles represent genovariants GII.7[P7] identified in the  Sverdlovsk region in 2024»

Comments 7: Discussion

Page 14, Lines 402-407. Consider adding a brief note on the patients’ clinical course and exploring whether the observed amino acid substitutions (e.g., in GII.4[P16]) could be related to clinical manifestations. If data on disease severity are available, incorporating this information could further highlight the potential clinical relevance of the genetic findings.

Response:

The study sample lacks information on the severity of the clinical course. Unfortunately, we cannot interpret the impact of mutations in VP1 antigenic determinants on disease severity.

Comments 8: The manuscript would be strengthened by the inclusion of a brief discussion of the study’s limitations.

Response:

One of this study’s limitation is the choice of method for genotyping. Different laboratories may use varying methods and protocols for norovirus genotyping, leading to significant differences in results and reducing the overall reliability and comparability of genotyping data. Another limitation in our study is the number norovirus strains that were included in the NGS analysis. We choose five GII.4 strains and five GII.7 strains as the most prevalent HuNoV genotype in our region. We could not include more HuNoV strains due to financial limitations and the need to achieve a better genome coverage by NGS. Due to same reasons, we did not include in the full-genome analysis the less prevalent circulating genotypes. 

The brief discussion of the study’s limitations has been added (lines 429-437):

«One of the main limitation in HuNoV genotyping is the choice of method. Different laboratories may use varying methods and protocols for, leading to significant differences in results and reducing the overall reliability and comparability of genotyping data. The main limitation in our study is the number of included norovirus strains in the NGS analysis. We choose five GII.4 strains and five GII.7 strains, as the most prevalent and representative HuNoV genotypes in our region. We could not include more HuNoV strains due to financial limitations and the need to achieve a better genome coverage by NGS. Due to same reasons, we did not include the less prevalent circulating genotypes despite the vast diversity of HuNoV genotypes in our results.»

On behalf of all the authors, thank you for your kind consideration!

Roman O. Bykov, Postgraduate student of the Federal Budgetary Institution of Science State Scientific Center of Virology and Biotechnology "Vector" Junior Research Assistant in the Laboratory of Enteric Viral Infections. Ekaterinburg Research Institute of Viral Infections, Federal Budgetary Institution of Science «Federal Scientific Research Institute of Viral Infections «Virome» Federal Service for Surveillance on Consumer Rights Protection and Human Wellbeing Rospotrebnadzor, 620030, Ekaterinburg, Russian Federation +7(982) 690-69-64
